# Accessory genes define species-specific routes to antibiotic resistance

Lucy Dillon[1] , Nicholas J Dimonaco[1,2,3] , Christopher J Creevey[1]

**A deeper understanding of the relationship between the antimicrobial resistance (AMR) gene carriage and phenotype is necessary to develop effective response strategies against this global burden. AMR phenotype is often a result of multi-gene interactions; therefore, we need approaches that go beyond current simple AMR gene identification tools. Machine-learning (ML) methods may meet this challenge and allow the development of rapid computational approaches for AMR phenotype classification. To examine this, we applied multiple ML techniques to 16,950 bacterial genomes across 28 genera, with corresponding MICs for 23 antibiotics with the aim of training models to accurately determine the AMR phenotype from sequenced genomes. This resulted in a >1.5-fold increase in AMR phenotype prediction accuracy over AMR gene identification alone. Furthermore, we revealed 528 unique (often species-specific) genomic routes to antibiotic resistance, including genes not previously linked to the AMR phenotype. Our study demonstrates the utility of ML in predicting AMR phenotypes across diverse clinically relevant organisms and antibiotics. This research proposes a rapid computational method to support laboratory-based identification of the AMR phenotype in pathogens.**

## Introduction

The overuse and misuse of antibiotics has escalated the rate at which many bacteria have evolved and acquired resistance to multiple antibiotics (Byrne et al, 2019; Sabino et al, 2019), including last-resort treatments (Andrade et al, 2021). This has led to growing prevalence of antimicrobial-resistant infections worldwide (Kwon & Powderly, 2021), which can be challenging to treat (Brauner et al, 2016). This has caused antimicrobial resistance (AMR) to become an increasing burden on society from a global health, agricultural, and financial perspective (Huws et al, 2018; VanOeffelen et al, 2021; WHO, 2021). If the rate of AMR continues as projected, it is estimated that by the year 2050, there will be >10 million deaths annually as a result of AMR-related infections (Kumar et al, 2021).

The AMR phenotype is typically distinguished through laboratory-based approaches such as broth microdilution, E-tests, or disc diffusion assays (van Belkum et al, 2020). However, it typically takes a minimum of 2–4 d to culture the bacteria and then complete the test (Berglund et al, 2019). More rapid testing is available through automated instruments for antibiotic susceptibility testing, such as commercial automated antimicrobial susceptibility tests (i.e., VITEK 2 system and MicroScan WalkAway) (van Belkum et al, 2020; Khan et al, 2021) or isothermal microcalorimetry to accurately determine MIC values (i.e., Symcel) (Tellapragada et al, 2020), which is often used in clinical environments. However, although these assays are usually a good estimate of the AMR phenotype in culture, this does not always translate to clinical settings. This is further complicated by the difficulty in culturing many organisms, especially when assessing species directly from microbiome samples (Raymond et al, 2019). Besides the importance of understanding the role of the AMR phenotype in microbiomes from an AMR reservoir perspective (Sabino et al, 2019), it also has the potential to reveal the mechanisms underpinning AMR-driven dysbiosis (Miyoshi et al, 2017) within humans and animals and potentially aid in preventing disease while concomitantly slowing the spread of AMR.

Recently, computational methods to identify AMR-causing genes in genomic data have become widely available (McArthur et al, 2013; Hunt et al, 2017; Bortolaia et al, 2020) and are often used to assess the potential antibiotic resistance phenotype of an organism (Sabino et al, 2019) or even entire microbiomes (Zaheer et al, 2019). These AMR gene identification tools run relatively quickly, especially compared with laboratory-based assays. Still, different tools can provide varying results (Feldgarden et al, 2019), likely driven by differences in the databases and varying methods to detect AMR genes (Doster et al, 2020). Very often microbiomes harbour AMR genes even when antibiotic usage is absent (Zhou et al, 2020; Gupta et al, 2021; Ma et al, 2022). Why these bacteria harbour AMR genes within the microbiome is unclear, and although several AMR genes have been reported to have alternative functions, such as transporters (Cudkowicz & Schuldiner, 2019), this is not the case for all.

[1]School of Biological Sciences, Queen's University Belfast, Belfast, UK   [2]Department of Medicine, McMaster University, Hamilton, Ontario, Canada   [3]Farncombe Family Digestive Health Research Institute, McMaster University, Hamilton, Canada

Correspondence: ldillon05@qub.ac.uk

Most importantly, the use of AMR gene identification tools to predict the AMR phenotype represents what in many cases is likely to be an oversimplification of the mechanisms underpinning AMR: that a single gene or mutation is solely responsible for the presentation of the AMR phenotype. In this situation, other non–AMR-associated genes may be required to confer resistance (or susceptibility) of an organism to an antibiotic (de Korne-Elenbaas et al, 2022). In the rest of this study, we refer to these non-classical AMR genes that are important to the presentation of the AMR phenotype as AMR "accessory" genes.

Previous studies attempting to use machine learning as a way of predicting the AMR phenotype from the genotype (Nguyen et al, 2019, 2020) have been limited by only studying a specific species, and/or using a single antibiotic (Nguyen et al, 2018; Macesic et al, 2020; Wang et al, 2022; Yasir et al, 2022) or non-interpretable methods such as neural networks (Avershina et al, 2021), thereby limiting the ability to understand the biological processes involved. Using a more interpretable method such as decision trees, as applied to a wide range of taxa and antibiotics, has the potential to provide a unique biological understanding of antibiotic resistance and allow the identification of accessory gene involvement in alternative "routes" to phenotypic resistance.

To address this, we set out to identify whether careful curation of data combined with the use of interpretable ML methods could elucidate the role of "accessory" genes in the presentation of an AMR phenotype.

Our underlying hypothesis is that focusing solely on classic AMR genes misses vital information needed to evaluate AMR phenotypes accurately. We address this through the application of multiple machine-learning (ML) models to a dataset of 16,950 genomes from microbial isolates representing 28 different genera with 1.9 million corresponding laboratory-determined MICs for 79 different antibiotics. These data were filtered by matching to EUCAST breakpoints (EUCAST, 2021) to ensure more balanced datasets according to the AMR phenotype. The filtering resulted in 5,990 genomes across 19 genera, with 47,711 EUCAST classified MICs for subsequent analysis (28,480 resistant and 19,231 susceptible MICs) of 23 antibiotics. We then elucidate the genomic routes (combinations of genes present and/or absent in genomes) involved in phenotypic antimicrobial resistance with the aim of allowing for the development of rapid determination of the AMR phenotype from genomes or even whole microbiomes.

## Results

### Machine-learning approaches vastly improve AMR phenotype prediction from the AMR genotype

Within this study, we analysed several techniques for predicting the AMR phenotype from genomic data, including a naïve analysis of AMR genes, logistic regression of AMR genes, J48 decision tree, random forest, support vector machine (SVM), and logistic model tree (LMT) models.

Even though AMR gene identification tools are designed to identify the presence of AMR genes in genomic data, their results are frequently used to directly infer the AMR phenotype (Bortolaia et al, 2020; Tan et al, 2020; Florensa et al, 2022; Verschuuren et al,

2022). We examined the accuracy of predicting the AMR phenotype solely based on the presence/absence of known AMR genes for 23 antibiotics and 16,950 genomes, from organisms with laboratory-derived MIC data. This naïve model assumed an antibiotic-resistant phenotype when an AMR gene, which targets a particular antibiotic (as defined in the CARD), was found in a genome.

The average prediction accuracy of this naïve (Resistance Gene Identifier [RGI]-specific) analysis (as defined by the number of genomes correctly predicted to be susceptible or resistant to an antibiotic divided by the total number of genomes tested) was 57.6% and ranged from 3.5% (clindamycin) to 100% (moxifloxacin) (Fig 1A). Clindamycin had quite a poor ratio of susceptible to resistant genomes (273:10) in comparison with moxifloxacin, which, while having far fewer genomes for training, had a better ratio of susceptible to resistant genomes (4:10), which may have led to the higher accuracy observed. The precision and recall were calculated using a confusion matrix (Tables S1 and S2). The average prediction precision was 56.2% and ranged from 46.3% (fosfomycin) to 100.0% (moxifloxacin) (Tables S2 and S3). The average prediction recall for all 23 antibiotics was 61.2% and ranged from 24.6% (ertapenem) to 100.0% (moxifloxacin).

When logistic regression approaches were applied, the resulting models of the RGI genes had an average accuracy of 73.9% and ranged from 50.96% (erythromycin) to 97.44% (amoxicillin). However, >50% of the models only predicted one phenotype (either only susceptible or only resistant predictions), resulting in an average recall of 52.3% (ranging from 48.5% [doripenem] to 75.0% [amoxicillin]) and the average precision of 53.6% (ranging from 31.5% [doripenem] to 74.5% [erythromycin]) (Tables S2 and S4).

When a decision tree approach (using the WEKA J48 model) was applied to the RGI-specific dataset, the resulting models were highly accurate in predicting the correct AMR phenotype: 10-fold cross-validation resulted in an average accuracy of 91.1% ranging from 74.85% (tigecycline) to 100% (moxifloxacin) (Fig 1A and Tables S4 and S5). The average recall of the RGI-specific decision tree models was 76.8% (ranging from 50.0% for amoxicillin, aztreonam, clindamycin, colistin, fosfomycin, and nitrofurantoin to 100.0% for moxifloxacin) (Fig 1C). The average precision was 86.2% (ranging from 43.0% for colistin to 100.0% for moxifloxacin) (Fig 1B). Furthermore, the traversal of the resulting decision trees indicated different genomic routes to resistance and susceptibility (see Fig 2), highlighting the importance of both the presence and the absence of multiple genes in predicting the AMR phenotype from genomic data.

The average accuracy of the J48 model (91.0%) was comparable to that of the random forest (92.0%), SVM (86.3%), and LMT (92.2%) models (Table S6 and Fig S1). The decision tree models had the advantage over the other models of allowing biological interpretation of the genes driving the AMR phenotype/genotype relationship. For this reason, we focused further analysis on the decision tree models.

### Model accuracy is not reliant on specific taxonomy

To investigate the ability of the decision tree models to predict the AMR phenotype for groups of organisms that were not included in the training data, for each antibiotic model, we

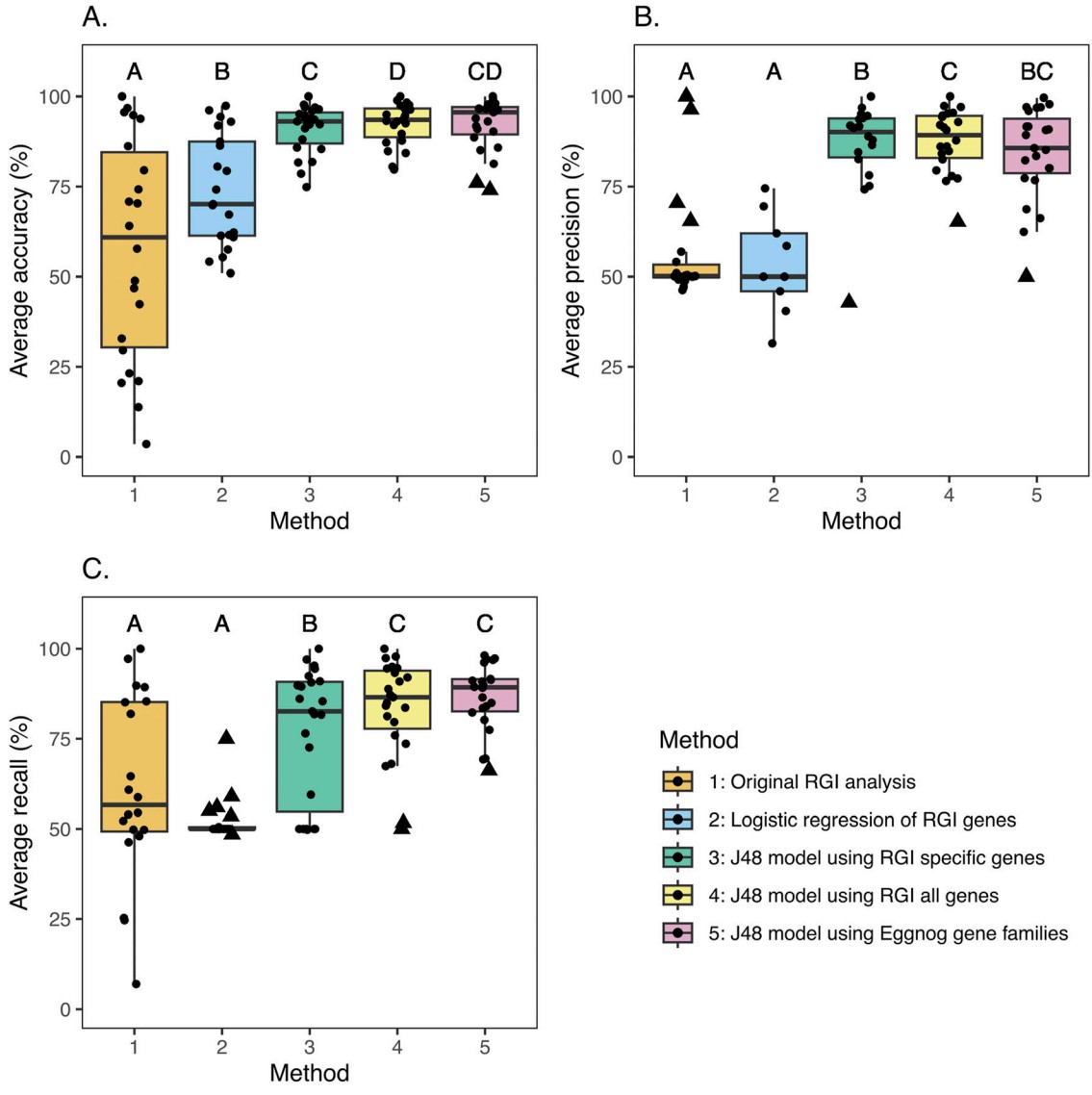

**Figure 1. Comparison of techniques used predict AMR phenotype from genomic data.**
**(A, B, C)** Model average accuracy (A), average precision (B), and average recall (C). The boxplots represent the following methods used in this study to predict the AMR phenotype in the following order: naïve RGI analyses (orange), logistic regression using the RGI data (blue), J48 decision trees using RGI genes specific to the antibiotic (green), J48 decision trees using all RGI genes regardless of the antibiotic model (yellow), and J48 decision trees using eggNOG gene families (pink). The statistical significances are the result of a pairwise Wilcoxon signed-rank test adjusted for multiple testing using the Benjamini–Hochberg method (q < 0.05). No significant difference between distributions is indicated by a shared letter above their respective boxplot (see Table S7 for more details). Outliers are represented with a triangle-shaped point.

generated multiple sub-datasets where for each we excluded all genomes (and MIC data) from a selected genus from the training data and regenerated the model. The excluded genus and associated MIC data were then used to test the accuracy of the regenerated model for predicting the AMR phenotype across taxonomic groups. Overall, the average accuracy of models to predict the AMR phenotype for a genus that was excluded from the training data was 80.3% (ranging from 0% to 100%) (Table S3). This varied depending on the taxa excluded; for instance, when 152 *Streptococcus* genomes and MICs were excluded from the meropenem data, the resulting models were able to predict 100% of *Streptococcus* AMR phenotypes. However, in the ampicillin

model, *Salmonella* phenotypes were not predicted well (34% accuracy) when all 1,048 *Salmonella* genomes and MICs were left out of the training data. This variation in performance across taxa may be due to an uneven distribution of susceptible versus resistant data in some of the training data (for instance, for testing the *Streptococcus* AMR phenotype, there were 383 resistant genomes versus 1,807 susceptible genomes, compared with 2,364 resistant genomes versus 13 susceptible genomes for testing the *Salmonella* AMR phenotype). *Klebsiella* had the most genomes for each antibiotic model tested (at least 1,248 genomes in each model), yet models that excluded them still performed relatively well (with an average accuracy of 84.4%). However,

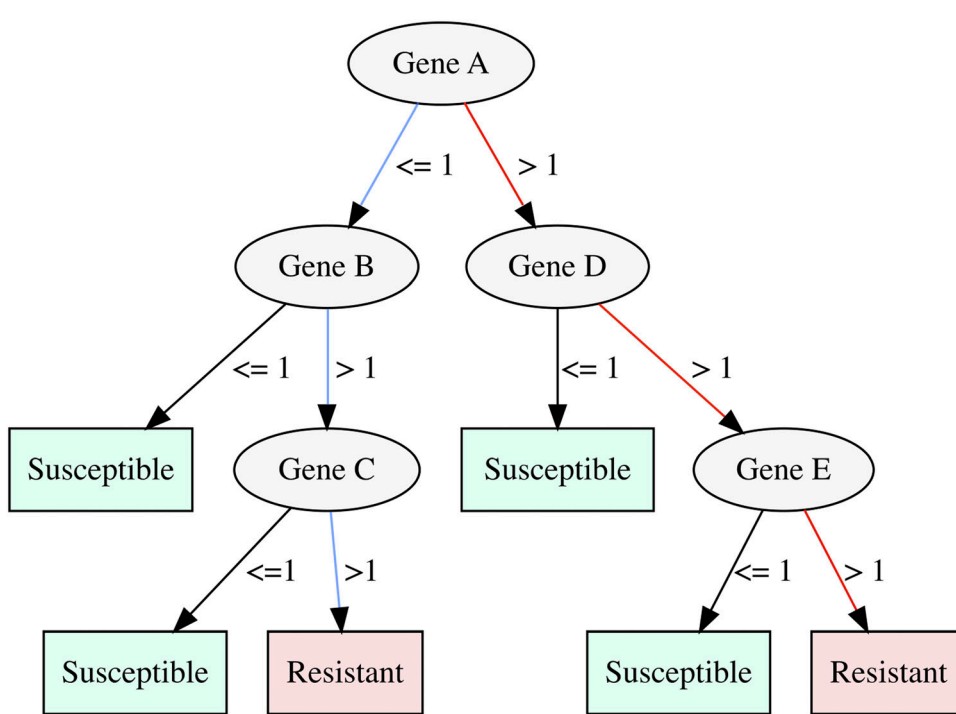

**Figure 2. Example of a decision tree with two routes to resistance, indicated by the red and blue lines.**
For example, in the red route, if more than one copy of Gene A, D, and E is present, the genome will be resistant, but if one of those genes is not present (i.e., Gene E), the organism will be susceptible.

sometimes when the genus with the second largest number of genomes for an antibiotic was excluded from the training data, the resulting models predicted their AMR phenotype poorly. For example, with ampicillin when the second largest genus *Salmonella* (i.e., 1,048 *Salmonella* genomes versus 1,923 *Klebsiella* genomes) was excluded, the resulting model had an accuracy of only 34.3% when predicting the *Salmonella* AMR phenotypes. For Ciprofloxacin, however, when *Salmonella* (which was the second largest genus in this model after *Klebsiella*) was excluded from the data, the resulting models had an accuracy of 97.7% when predicting the *Salmonella* AMR phenotypes. This may be due to some genera being more closely related than others, such as *Klebsiella* and *Escherichia* compared with *Neisseria*. However, this does not account for all cases and higher accuracy of predicting the AMR phenotype using models trained on more distantly related organisms may be driven by other factors, such as convergent evolution in species exposed to similar conditions or horizontal gene transfer, resulting in similar resistance mechanisms between the species. Investigation of the decision tree models supports this theory: for Ceftriaxone, ciprofloxacin, and gentamicin where the models performed well in predicting the *Salmonella* AMR phenotype when *Salmonella* was excluded from the training data, the tree traversals identified that each route to phenotype resistance was supported by data from multiple genera (Fig S6). However, for ampicillin where the models performed poorly in predicting *Salmonella* AMR phenotypes when the *Salmonella* data were excluded from the training data, the tree traversals showed that most of the routes to AMR were dominated by a single genus. Therefore, when *Salmonella* was excluded from the training data, the routes to resistance for *Salmonella* were lost from the model, resulting in low accuracy.

This insight into the mechanics of the models is only possible because of the use of an interpretable ML model, which not only highlights the often species-specific routes to AMR that exist, but also hints at those routes to resistance that are readily shared between taxa and may be important to monitor for early-warning surveillance programmes.

**ML models identify putative additional antibiotic targets of AMR genes**

To investigate the role of AMR genes in antibiotic resistance to which they are not indicated in the CARD, we generated decision trees, which included all AMR genes regardless of the antibiotic target listed in the CARD. This resulted in 17 antibiotic models improving in accuracy, and across all models, a significant increase in accuracy was observed compared with the models using only the AMR genes specific to the antibiotic that is listed in CARD (Wilcoxon's signed-rank test [q = 8.27 × 10$^{-4}$] [Table S7]). Results of investigations of model prediction accuracies for one AMR phenotype over the other can be found in confusion matrices in Tables S1, S5, S8, S9, and S10. The average accuracy of the models using all AMR genes regardless of the antibiotic to which they were indicated to provide resistance was 92.5% (ranging from 79.7% [tigecycline] to 100.0% [moxifloxacin]). The average recall and precision were 83.5% (ranging from 50.0% [fosfomycin] to 100.0% [moxifloxacin]) and 87.5% (ranging from 65.0% [nitrofurantoin] to 100.0% [moxifloxacin]), respectively (Fig 1B and C and Table S2). A significant increase in average recall and precision was also observed (recall, q = 4.39 × 10$^{-4}$; and precision, q = 0.04) (Table S7). This suggests that some AMR genes may have additional antibiotic targets not annotated in the databases. One example of this can be seen in the

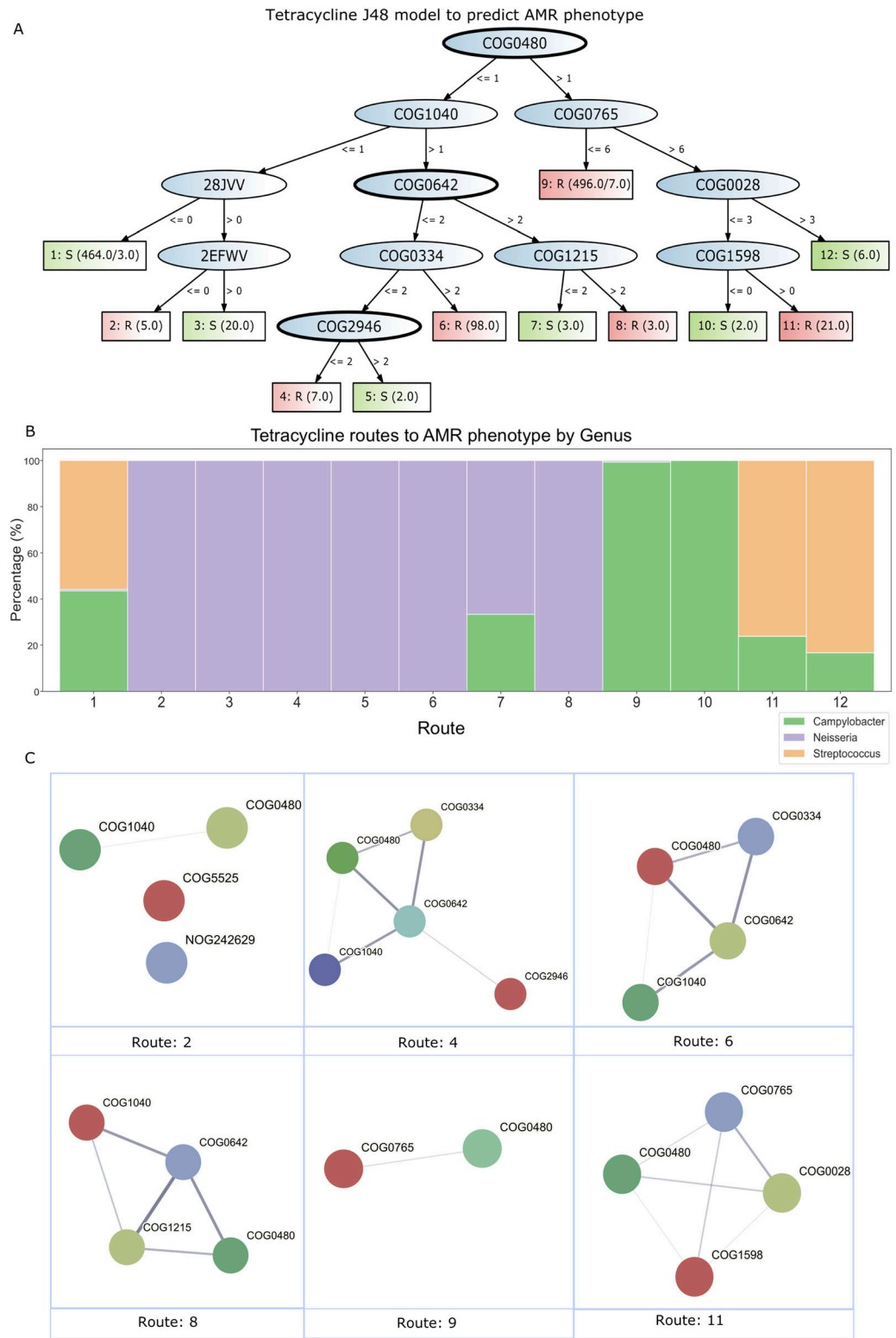

**Figure 3. Predicting tetracycline resistance using eggNOG gene family copy number or absence.**
**(A)** J48 decision tree model to predict the tetracycline AMR phenotype. RGI-associated gene families have been highlighted with a thick black outline. COG0480 relates to gene *tet(44)*, COG0642 relates to gene *adeS*, and COG2946 relates to gene *tetU*. The decision trees have numbers in the phenotype boxes to represent the number of genomes. This may include two numbers in some cases, the first number indicates the total number of genomes, and the second number is the number of incorrectly classified genomes.* **(B)** Stacked bar chart showing the routes to susceptibility and resistance for tetracycline. This is a genus-level analysis; the species, family, order, class, and phylum analysis can be found in Fig S6. The route numbers relate to the numbers on the decision tree (part (A)). Note: route 9 is not for 100% *Campylobacter*, but

gentamicin RGI-all model, which shows the presence of >1 *TEM-185* gene confers resistance to gentamicin (Fig S2). This gene is not indicated to confer resistance to aminoglycoside antibiotics in the CARD.

### Accessory genes have a key role in AMR phenotype prediction

To see whether this observation extended to non-classical AMR genes, decision trees were generated for the 23 antibiotics using eggNOG gene family functional profiles generated for all 16,950 genomes. The average accuracy of these models was 92.2% (ranging from 74.0% [tigecycline] to 100.0% [moxifloxacin]). In the comparison of the eggNOG models with the RGI models, the mean value was 0.3% higher for RGI-all analysis (92.5%) (Tables S4 and S10). The difference between the RGI decision tree models and eggNOG gene families was not significant overall (RGI-specific genes versus eggNOG, q = 3.66 × 10$^{-1}$; RGI-all-gene models versus eggNOG, q = 2.49 × 10$^{-1}$) (Table S7). Overall, the inclusion of AMR accessory genes in the models did not reduce the accuracy, precision, or recall compared with the AMR gene–based decision trees; however, their inclusion allowed the identification of putative accessory genes involved in species-specific routes to antibiotic resistance (Fig 1A–C). Using the eggNOG decision trees, we identified an additional 675 gene families across all 23 antibiotic models, which are not in the RGI database but were indicated as linked to the AMR phenotype.

### Decision trees identify species-specific biological routes to resistance

The use of decision trees allowed biological interpretation of routes to resistance and susceptibility predicted by the models (428 susceptible routes and 528 resistant routes across all the eggNOG models, Figs 3, S2, S3, and S4 and Table S11). In some cases, putative novel roles of known AMR genes were identified; for instance, in the eggNOG-based amikacin model, COG0050 is matched to a multi-drug–resistant gene (*Escherichia coli acrA*—as named in the CARD), but this AMR gene is not involving in aminoglycoside resistance, suggesting that this gene may have additional targets. The routes also highlighted not only the importance of the presence (and number of copies) of key AMR and accessory genes to antibiotic susceptibility and/or resistance of an organism but critically also the importance of the absence of certain genes to these phenotypes. It is possible that the inclusion of information about genes not present in a genome and/or the co-occurrence of particular genes may have played an important role in the observed increase in model accuracy over the naïve approaches that simply use the presence of a known AMR gene to indicate the AMR phenotype (Fig 1). Although this highlights key genes involved in the AMR phenotype that are not classic AMR genes (Fig S1), reassuringly, when RGI genes were matched to the

gene families in the decision tree models, we found that most of the models also contained known AMR (RGI) gene families. An interesting insight that was only possible because of the use of decision trees was the sometimes large number of different resistance routes possible for a single antibiotic. For example, the tetracycline decision tree model using eggNOG gene families identified six different routes to resistance across all genomes analysed. As can be seen in Fig 3A, the leaves of decision trees have values for each phenotype representing the number of genomes in the training set that take that route to resistance (in the case in which there are two numbers, the first is the total number of genomes and the second is the number of incorrectly classified genomes). The most common route to tetracycline resistance involves COG0480 and COG0765 (both positive and negative involvement with resistance). The gene family COG0480 is a known key gene family involved in tetracycline resistance (*tet(44)*); however, Fig 3A also shows that this gene does not have to be present for an organism to be resistant.

The fact that known AMR genes (RGI gene families) did not dominate the trees and routes to resistance suggests that accessory genes have an important role in the AMR phenotype. For example, the eggNOG-based decision tree using the tetracycline phenotypic data had three known AMR gene families *tet(44)*, *tetU*, and *adeS* (COG0480, COG2946, and COG0642, respectively), yet their presence did not always guarantee resistance to tetracycline (Fig 3A). Therefore, the presence or absence of certain accessory genes is necessary to confer the resistant phenotype.

STRING (version 11.5) (Snel et al, 2000) was used to identify predictions of putative protein–protein interactions between genes within each route to resistance under the hypothesis that if these routes and especially the involvement of non-classical AMR genes are valid, they should be enriched in predicted protein–protein interactions. Of the 23 models, 18 contained eggNOG gene families, which were predicted to have protein–protein interactions, predicted from co-occurrence and co-expression in STRING. The tetracycline decision tree using eggNOG gene families showed that 63.6% of the gene families had predicted protein–protein interactions (Fig S5). In addition to the evidence-based predicted protein–protein interactions across each decision tree model, we analysed each route to resistance within the models. For each route to resistance, we determined the predicted protein–protein interactions using edges based on confidence (strength of data support) rather than based on evidence (indicates the type of interaction) (Fig 3C). The individual routes to resistance had an average of 1.2 predicted protein–protein interactions ranging from 0 to 7.6 (Table S11) (we included connections based on low confidence to provide further evidence that these putative connections that may not be well documented in the database may have a role in the AMR phenotype). This investigation also highlighted that many of the routes are taxonomically dependent. This was evident in the decision tree models using both AMR and

0.4% for *Neisseria*. **(C)** Protein–protein interactions between gene families for each route to resistance. The lines (edges) represent the protein–protein interactions from STRING, and the thicker the line, the higher the confidence (see Table S11 for details). See part (A) for details of each route (the route numbers 10/27 correspond to the numbers on the phenotype boxes in part (A)).* Note*: eggNOG gene families 28JVV and 2EFWV correspond to NOG242629 and COG5525 in the STRING database, respectively.

accessory genes. As an example, in Fig 3A we can see 12 distinct routes to the AMR phenotype (either susceptibility or resistance); routes 4–6 were only found in the *Neisseria* genus. Route 10 was only found in *Campylobacter,* and route 9 was dominated by *Campylobacter* (99.4% *Campylobacter*, 0.6% *Neisseria*) (Fig 3B). Although this is not the case for every route in the trees (i.e., route 1 is very mixed taxonomically), many of the branches in the trees could predict species of the genome analysed and the AMR phenotype. Interestingly, for those routes to resistance that were found in multiple different taxa (e.g., route 11 in Fig 3B), this might suggest routes to resistance that are more easily shared between species by mechanisms such as horizontal gene transfer. Additional antibiotic model routes for species–phylum can be found in Fig S6. Details on all other routes to resistance can be found in Table S11.

### Resistance routes to different antibiotics are distinct from each other but share some key genes

Using the decision trees, we can work out which combinations of genes are involved in resistance (see an example decision tree [Fig 2] for reference). Understanding which genes are key to resistance in particular antibiotics or shared across different antibiotics could help provide insight into novel approaches to combat AMR in future. To investigate whether there was an overlap in the genes involved in resistance to different antibiotics, we combined gene families present in all decision trees. The eggNOG gene family distribution across the different models was analysed resulting in 723 unique eggNOG gene families in total for all the models. Of these unique eggNOG gene families, only 48 were linked to RGI AMR genes (Table S12). The distribution of functional categories assigned to the eggNOG gene families (also known as COG categories) varied across all models, suggesting that resistance to different antibiotics was driven by distinct mechanisms (Fig S7).

To investigate possible connections between the networks of genes involved in resistance across different antibiotic models, we identified putative protein–protein interactions between all genes from all routes to resistance for all the antibiotic models using STRING. The resulting STRING network had 450 nodes and 10,786 edges. This included all evidence types in STRING at a medium level of confidence (0.4). We then reduced the network to include only those connections between genes where the genes were also found together in at least one decision tree model. The resulting reduced network had 247 nodes and 1,300 edges, which showed clear clusters linking gene families to specific antibiotic models (Fig 4). The network highlighted several key gene families, which are involved in many different antibiotic gene clusters (the larger nodes in Fig 4). This suggests these genes may have an important role in the AMR phenotype across multiple antibiotics. The largest node in the graph represents COG2367, which played a role in seven different antibiotic models. This gene is annotated as part of the defence mechanism group of eggNOG gene families, labelled as beta-lactamase. This may be expected as six of the models were built using the MICs of the beta-lactam antibiotics, but COG2367 was also part of the Nitrofurantoin model, which is distinct from the beta-lactam drug class. The nitrofurantoin

model predicts that it is possible to have up to four copies of COG2367 and still be susceptible (which according to the labels accounts for most of the genomes on this route through the decision tree); if more than four copies are present, then the model predicts that the organism can be resistant to nitrofurantoin. In the gene network, we can also see how different antibiotics have distinct genes, which are only associated with that particular antibiotic (indicated by the colours in Figs 4 and S8). This highlights those genes that are involved in unique routes to resistance to each antibiotic. Although the network in Fig 4 was generated without prior knowledge of the antibiotic drug classes that the genes were predicted to be associated with, we see clear sub-clusters of genes according to the antibiotic drug class. However, most interesting is how connected all these sub-clusters are even though some represent very distinct antibiotic drug classes including carbapenems and aminoglycosides (Fig S9). This suggests the possibility that certain key genes are important to the establishment of routes to resistance against multiple antibiotics leading to multi-drug resistance phenotypes and should be monitored in AMR surveillance initiatives.

## Discussion

Within this study, we have shown that machine learning can vastly improve the prediction of the AMR phenotype from genomic data and the inclusion of accessory genes with AMR genes in these analyses provides valuable biological insights into the routes to AMR and susceptibility.

Although AMR gene identification tools like RGI are not designed to identify the AMR phenotype but rather the AMR genotype, their results are often inferred as phenotypic resistance for genomes (Bortolaia et al, 2020; Tan et al, 2020; Florensa et al, 2022; Verschuuren et al, 2022) and metagenomes (Zaheer et al, 2018; Jankowski et al, 2022). Our results demonstrate how this naïve approach to AMR phenotype prediction results in an average accuracy of only 57.58%, which is comparable to a game of chance. The average precision was 56.2%, and the average recall was 61.2%, which highlights key flaws in using these tools to predict the AMR phenotype (Fig 1B and C).

Overall, although results from the logistic regression analysis were significantly better than those from RGI-only analysis (Fig 1), it is still underperformed, which could suggest there are not enough data to make an accurate model, there is not a strong enough link between the phenotype and the genes to be able to classify accurately using this approach, or the model is not complex enough. However, decision trees show over 17% increase (statistically significant, q = 6.03 × 10$^{-5}$, Table S7) in accuracy compared with logistic regression, suggesting that the poor performance of logistic regression may be due to the limitation of using only presence/absence information rather than the number of copies of genes that the decision trees are capable of using it.

The decision tree results have shown that both the presence (including the number of copies) and the absence of different gene families are key in the accurate prediction of the AMR phenotype. Biologically, this makes sense as we know that genes perform their

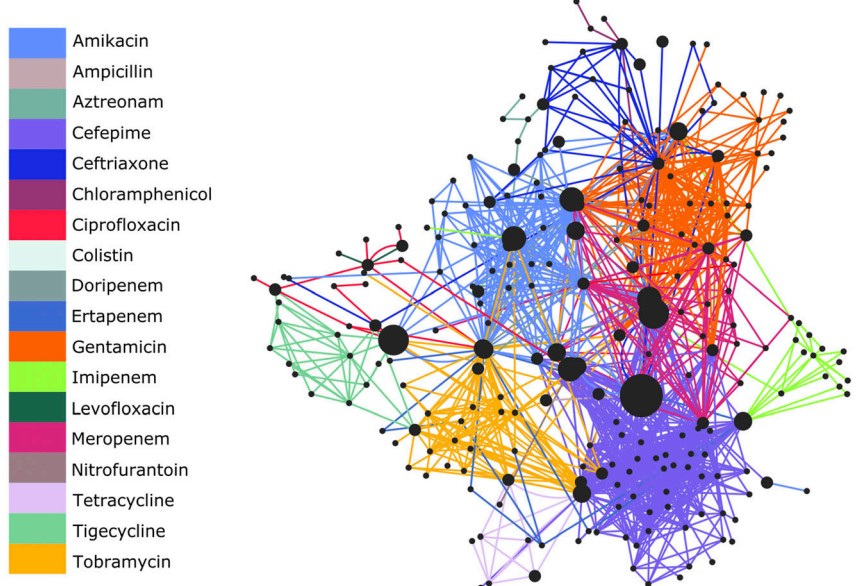

**Figure 4. Gene network of all unique eggNOG gene families across all antibiotic eggNOG models.**
Nodes are the different eggNOG gene families from all routes to the antibiotic AMR phenotype identified from the decision tree models, and edges are protein–protein interactions predicted between eggNOG gene families by STRING. Only those predicted protein–protein interactions between genes that were also found together in at least one decision tree model are shown. The edge colour corresponds to the antibiotic model that the eggNOG gene family pair is present in. The node size is proportional to the number of models the eggNOG gene family is present in.

Legend: Amikacin, Ampicillin, Aztreonam, Cefepime, Ceftriaxone, Chloramphenicol, Ciprofloxacin, Colistin, Doripenem, Ertapenem, Gentamicin, Imipenem, Levofloxacin, Meropenem, Nitrofurantoin, Tetracycline, Tigecycline, Tobramycin

function most often as an ensemble with other genes. Supporting this, the decision trees show that even when a known AMR gene is present, it does not necessarily mean that the organism is resistant (Figs S2, S3, and S4). Interestingly, the decision tree models, which included all known AMR genes (and not just those thought to be involved in resistance to the antibiotic being examined: RGI-all models), showed an increase in accuracy compared with the models that were generated using only those AMR genes known to provide resistance to the specific antibiotic. This suggests that some of the AMR genes within the CARD may be involved in providing resistance to a broader range of antibiotics than what is annotated. However, AMR genes may need additional genes present to confer resistance to particular antibiotics, which are not identified in any commonly used AMR gene identification tool.

The use of eggNOG gene families has shown the importance of accessory genes in the role of the AMR phenotype. Accessory genes are generally ignored when determining the AMR phenotype of an organism when using computational techniques to predict the AMR phenotype. Therefore, studies that rely on AMR gene identification tools to determine resistance could be misleading as the full genomic picture is not described. All the eggNOG decision tree models are dominated by non-AMR genes (Fig S4) and show in which contexts the presence of an AMR gene does not guarantee resistance to a particular antibiotic. Almost 30,000 gene families were used to train the eggNOG-based J48 models, in comparison with the 1,424 RGI AMR genes used to train the RGI-all gene J48 models. The accuracy, precision, and recall did not differ significantly between these models, suggesting that the inclusion of the extra non-classical AMR genes did not affect the ability of the J48 model to accurately predict the AMR phenotype. However, their inclusion did allow a better biological understanding of the routes to AMR used by different species.

Analysis of the protein–protein interaction networks of individual routes to resistance showed high levels of predicted interactions, which would be expected if all the genes identified were involved in a shared process, such as establishing resistance to an antibiotic. Yet, when all the protein–protein interactions were examined across all the antibiotic models, clear connections between them were evident, suggesting there are key non-AMR annotated genes involved across many antibiotic resistance mechanisms (Fig 4).

Identifying taxonomically dependent routes to resistance within the decision trees highlights key genes that could be targeted for particular pathogens. Conversely, routes to resistance with multiple taxa involved could suggest a route that is more transmissible between species and may pose a greater risk in clinical settings. Overall, the routes to resistance provide unique biological insights into AMR mechanisms, many of which are understudied.

Although more data and improved models are needed to provide accurate predictions for wider taxonomic diversity than available for this study, it is clear that the AMR phenotype cannot be clearly explained by the use of AMR gene identification tools alone, and a more complex approach, such as the one implemented in machine learning, is required. Although we concentrated on the results of the J48 decision tree models as they provide more biological insights than other machine-learning models such as random forest, SVM, and LMT, we found comparable levels of phenotype prediction accuracy using these other ML approaches. Further work needs to be done in this area to develop biologically interpretable machine-learning models for the important insights that they can provide about the AMR phenotype.

## Summary

AMR gene identification tools that identify the presence or absence of known AMR genes are not designed to be used to predict the AMR phenotype despite their wide use for this purpose. Therefore, it is unsurprising that multiple studies have reported their inability to

accurately predict the AMR phenotype of many organisms. However, in this study, we have demonstrated that careful curation of data and application of even basic ML models can overcome these limitations and accurately predict the AMR phenotype for a wide range of clinically relevant taxon and antibiotic combinations. In addition, we demonstrate that the incorporation of all genes from a genome (and not just the known AMR genes) allows the identification of species-specific routes to antibiotic resistance, involving genes not previously identified as involved in this process. Lastly, we have shown that the inclusion of factors, such as the number of copies of a gene family, and their absence are important key factors that need to be considered when predicting the AMR phenotype from genomes.

The models generated in this study are the first step towards tools that could be used in clinical environments to aid laboratory-based diagnostics. Although the species-specific nature of many of the routes to phenotypic resistance identified suggests that the current models would be best applied to clinically relevant isolates closely related to the taxa used to train the models, their taxonomic specificity could also allow identification of recently horizontally transferred resistance mechanisms from these species into a wider range of taxa. Lastly, the accessory genes identified as involved in the AMR phenotype could also provide novel drug targets to combat AMR.

Overall, this study demonstrates that complex taxonomically dependent genetic components drive the AMR phenotype in a wide range of species and that machine-learning methods have the potential to provide rapid computational methods to support laboratory-based identification of the AMR phenotype in pathogens.

A call for data: If you would like to be involved in improving these models by contributing genomes with corresponding MIC (broth microdilution) data, please contact us at: ldillon05@qub.ac.uk

# Materials and Methods

All scripts and files mentioned in the text can be found at https://github.com/LucyDillon/AMR_ML_paper/tree/main. This includes all bash scripts to analyse the data and details of how gene counts for RGI and eggNOG gene families were calculated.

Supplementary files and additional data can be found at: https://osf.io/CJ4BQ/

### Data for analysis

Using the PATRIC command line interface (version 1.034—now known as BV-BRC) (Davis et al, 2020; Olson et al, 2022), 16,950 bacterial genomes from isolates of known taxonomy with 1,249,188 corresponding laboratory-determined MIC values were sourced. The genomes used in this study can be found using a wget command in the bash script: PATRIC_genomes.sh using the input: genome_ids.txt. For each genome, the AMR genotype was determined using the RGI tool v5.1.1 (Jia et al, 2017) with the CARD v3.1.1 (McArthur et al, 2013) using the default parameters and the whole

genome sequence from the genome as input. The CARD database includes acquired resistance and resistance due to mutations.

Each predicted AMR gene in each genome was then associated with the specific antibiotic(s) to which it was listed as conferring resistance to using the information in the CARD. The MIC values were categorized into "Susceptible" or "Resistant" using EUCAST breakpoints (Jan 2021 release) (EUCAST, 2021), which are taxonomic-specific MIC values that can differ between species. The MICs were categorized into the respective EUCAST breakpoints using custom Python scripts (OG_RGI_analysis.py, Logistic_regression_RGI.py, RGI_specific_analysis.py, RGI_all_analysis.py, and Eggnog_analysis.py). Any MIC values that fell outside the EUCAST definition of "Susceptible" or "Resistant" for any specific species were removed from the analysis. In the case that a genome had >1 MIC values for the same antibiotic, the average was calculated and then compared with the EUCAST breakpoints. This resulted in 5,990 genomes across 19 genera, with 47,711 EUCAST classified MICs for subsequent analysis (28,480 resistant and 19,231 susceptible MICs). Details of the number of each genus for each antibiotic model can be found in Table S13.

### Analysis of the AMR genotype-to-phenotype relationship

In this study, we used several techniques to further understand the relationship between the AMR genotype and the AMR phenotype. The models used are binary classifiers (either classifying as susceptible or resistant), which, although makes for a simpler model, excludes the use of intermediate resistance or more complex conditions such as persistence or tolerance. To predict the AMR genes present within each genome, we used RGI, a commonly used AMR gene identification tool. We evaluated four phenotype prediction approaches using linked laboratory-determined resistance/susceptibility profiles against a range of antibiotics. We first tested a naïve prediction of the AMR phenotype using the presence/absence of AMR genes and the antibiotics to which the genes were listed as conferring resistance in the RGI database. Secondly, we tested the application of a basic logistic regression model to the AMR gene presence/absence data. Finally, we tested the application of four machine-learning approaches to predict the AMR phenotype using gene counts of known AMR genes with and without gene counts of all other functionally annotated genes in the genomes (eggNOG gene families). Each of these approaches (further outlined below) was independently applied to the prediction of resistance to 23 different antibiotics for which relevant MICs were available.

### Naïve prediction of the AMR phenotype

Although RGI and other AMR gene identification tools do not claim to be able to infer the AMR phenotype, the presence of an AMR gene is often used to designate whether a genome is susceptible or resistant (Bortolaia et al, 2020; Tan et al, 2020). Therefore, the presence of an RGI-annotated AMR gene was used as an indicator of resistance to the antibiotic(s) to which the gene was labelled as resistant in the CARD. Precision, recall, and accuracy of both susceptible and resistant phenotypes were calculated for this naïve

model using a custom Python script (OG RGI analysis.py) as a baseline to compare the subsequent models.

### Logistic regression prediction of the AMR phenotype

To evaluate the relationship between the AMR genotype and the AMR phenotype, a logistic regression model was used for each antibiotic (Fig 1) with a split of 3:1 between training and test datasets, respectively, using a custom Python script (Logistic_regression_RGI.py). This model evaluated how the presence or absence of specific AMR genes was related to the AMR phenotype. Model precision, recall, and accuracy of both susceptible and resistant phenotypes were calculated to evaluate the model efficacy and potential bias. The ratio of susceptible organisms to resistant organisms can help determine the likelihood of bias in the training data (Fig S10).

### Decision tree prediction of the AMR phenotype using only AMR genes

To understand how specific AMR genes may drive the relationship between the AMR phenotype and the AMR genotype, four machine-learning approaches were used. A custom Python script was used to convert the RGI gene counts into an Attribute-Relation File Format (ARFF) file (RGI_specific_analysis.py) using the csv2arff tool found at https://github.com/LucyDillon/CSV_2_arff. The J48 decision tree models were built as implemented in the WEKA machine-learning platform (version 3.8.5) (Witten et al, 2011). The J48 model is written in Java and is an adaptation of the landmark C4.5 algorithm. In this analysis, the model considers the number of copies or the absence of an AMR gene in relation to the AMR phenotype. J48 decision trees are used to classify each "instance," or genome, based on the provided labels (AMR gene count). The model evaluates the data overall and then splits the genomes based on their labels (one label-based decision for each split). Next, it repeats this process on the subsets of genomes until the model has reached a preset limit based on either model parameters, such as the minimum number of genomes per split, or a consensus split of the correct categorical variable, in this case, AMR phenotype (further details below).

This analysis was then repeated using the random forest, SVM (WEKA package: libsvm 3.25), and LMT models in WEKA to compare the efficacy of each machine-learning approach (Table S6).

Models for 23 different antibiotics were selected with respect to various data constraints. Each model is trained specific to a single antibiotic, and the genomes present in the model must have corresponding MIC values. For a model to be able to learn from the data and thus predict the correct AMR phenotype, the models had to have both susceptible and resistant organisms (Table S4). The proportion of organisms with a susceptible or resistant MIC value can be seen in Fig S10.

The J48 model was chosen for further analysis because of the interpretability of its decisions, hence providing the biological reasoning behind the predictions it made. The output of the J48 model is a human-readable tree of the decisions to partition the genomes (as resistant or susceptible) (Figs S2, S3, and S4). The default parameters were used for the WEKA J48 model; however, the parameters were first evaluated by a matrix comparing M (Minimum number of instances per leaf [decision]) and C values (Confidence value: the lower value indicates more pruning [removing less

informative leaves/decisions]) (Table S14). There was no difference in eight of the antibiotic models using the different parameters, and the rest of the models had minor differences. The most accurate C value could be found to be 0.25 or 0.5 for 15 of 23 antibiotic models. The C value of 0.25 was selected as this level of tree pruning is recommended to not overfit the model or prune the tree too much and miss important information. The M value of 0.2 was selected as this is the default of the model and the other M values had very similar accuracy. The model accuracy was evaluated by 10-fold cross-validation. The individual fold results allowed the SE of the models to be calculated (Table S4).

To evaluate what factors may impact the models or improve model accuracy, the composition of AMR genes used to train the models was analysed. The models were originally trained using specific AMR genes for the antibiotic the model represented, for example, ampicillin-specific AMR genes to train the ampicillin model. The antibiotic target is defined in the CARD in which the genes are annotated to correspond to specific antibiotics. The models were then trained with all AMR genes present in the genomes regardless of which antibiotic model they were training (Tables S1 and S4 and Fig S11). A custom Python script was built to make the .arff files for this analysis (RGI_all_analysis.py).

### Investigating the role of taxonomy in decision tree model accuracy

To investigate the role of taxonomy in model accuracy, for each antibiotic model, one genus was excluded from the training data. The excluded genus was then used to test the model. This included each genus available for each antibiotic model (see Table S3 for details). This way, we can evaluate how the models may perform on a species that was not in the training set. We used a custom Python script to develop the .arff files (taxa_test_train_files.py). To process CSV files into the format required for WEKA (.arff), we created a simple tool to translate a .csv file into an .arff file. This code is freely available at https://github.com/LucyDillon/CSV_2_arff

### Analysis of accessory gene involvement in the AMR phenotype

To investigate the role of accessory genes in the AMR phenotype, the genomes from BV-BRC were analysed using Prodigal v2.6.3 (Hyatt et al, 2010), Diamond v0.9.24.125 (Buchfink et al, 2015), and eggNOG-mapper (version 2.1.6) to predict gene families (Cantalapiedra et al, 2021). All tools were used with default parameters. The least specific level of the eggNOG gene family (i.e., COG or NOG) was taken to get the most general result so that the gene families could be compared across different taxa. The number of genes present in a gene family, including their absence, was compared with the genome AMR phenotype using the same J48 model and parameters in WEKA, using a custom Python script to make the .arff files (Eggnog_analysis.py). A 10-fold cross-validation was used to evaluate the model's accuracy of predicting the AMR phenotype, from which the standard error was calculated.

We mapped the RGI AMR genes onto the eggNOG decision tree models by analysing the CARD with eggNOG-mapper. The eggNOG gene families reported were matched to the AMR genes and were then labelled as having a known AMR gene function in the models. Finally,

"routes to resistance" were identified in all of the resulting decision tree models by identifying all possible routes through the resulting trees that lead to a "Resistant" outcome, using Apply_Decision_Tree available at https://github.com/ChrisCreevey/apply_decision_tree. All gene families traversed to reach each resistance outcome on the decision trees were considered important to that resistance route (regardless of whether it is needed to be present or absent) and included in subsequent analyses of different routes to resistance.

### Investigating protein–protein interactions

Protein–protein interactions of the gene families within individual decision trees were investigated using the STRING protein–protein interaction database (version 11.5) (Snel et al, 2000). One protein sequence from each gene family was selected (the first sequence in the fasta file downloaded from eggNOG for each gene family) to represent that gene family (723 unique gene families in total) using the "Protein families," "COGs," and "multiple sequences" options for each individual antibiotic model in STRING. This analysis was used to highlight whether gene families within the same model or with the route to resistance were predicted to interact and therefore may have a role together in the AMR phenotype.

To find associations with the gene families across multiple models, STRING and Cytoscape (version 3.9.1) (Shannon et al, 2003) were used to analyse the data using the same options as above but including all gene families from all antibiotic models. To reduce the network to directly link to the decision trees, edges in the network were only retained if both gene families were present in the STRING protein–protein interactions (and therefore predicted to interact) and the pair was also present in at least one decision tree model. Details of this analysis can be found in the Cytoscape_analysis.md file.

The predicted protein–protein interaction network of each route to resistance was also produced using STRING but including only those gene families predicted for each individual route (allowing connections based on low confidence to provide further evidence that these putative connections that may not be well documented in the database may have a role in the AMR phenotype). To investigate whether routes to the AMR phenotype within the decision trees are taxonomically related, we traversed the decision trees to investigate which route each genome took for each model. This was performed using Apply_Decision_Tree using the same input genome .arff and dot files. DOT is a graph description language to visualize information, such as decision trees.

For all models, accuracy is defined by the sum of the true positives and true negatives divided by the sum of the total number of genomes (instances). Precision and recall of the models are calculated for both susceptible organisms and resistant organisms separately. This highlights whether a model is better at predicting one phenotype over another.

## Data Access Statement

All supplementary data and additional files can be found on the Open Science Framework at: https://osf.io/CJ4BQ/. For specific files, please email ldillon05@qub.ac.uk.

### Code availability

All codes used in this study can be found at the following links:
 All analysis scripts and files:
 https://github.com/LucyDillon/AMR_ML_paper
 A tool to convert a csv file to an .arff format file:
 https://github.com/LucyDillon/CSV_2_arff
 Apply_Decision_Tree tool (used for tree traversals):
 https://github.com/ChrisCreevey/apply_decision_tree

## Supplementary Information

## Acknowledgements

We acknowledge PhD funding from the Department for the Economy, Northern Ireland, to L Dillon. CJ Creevey wishes to acknowledge funding from UKRI (BB/R019185/1 and BB/R018464/1) and EU via Horizon 2020 (818368 MASTER and 101000213 Holoruminant). NJ Dimonaco wishes to acknowledge the Farncombe Family Digestive Health Research Institute (McMaster University) and a grant from the Weston Family Microbiome Initiative. This work was undertaken on Kelvin2, an EPSRC-funded Tier-2 High-Performance Computing Facility at Queen's University Belfast, UK.

### Author Contributions

L Dillon: conceptualization, software, formal analysis, investigation, visualization, methodology, and writing—original draft, review, and editing.
NJ Dimonaco: conceptualization, methodology, and writing—original draft, review, and editing.
CJ Creevey: conceptualization, data curation, software, supervision, investigation, visualization, methodology, and writing—original draft, review, and editing.

### Conflict of Interest Statement

The authors declare that they have no conflict of interest.

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
