## [Reviewer comments · Life Science Alliance]

Life Science Alliance

Accessory genes define species-specific routes to antibiotic resistance.

Lucy Dillon, Nicholas Dimonaco, and Christopher Creevey

DOI: <https://doi.org/10.26508/lsa.202302420>

Corresponding author(s): *Lucy Dillon, Queen's University Belfast*

Review Timeline:

Submission Date:	2023-10-06
Editorial Decision:	2023-11-27
Revision Received:	2023-12-22
Editorial Decision:	2023-12-28
Revision Received:	2023-12-29
Accepted:	2024-01-03

Scientific Editor: *Eric Sawey, PhD*

Transaction Report:

November 27, 2023

Re: Life Science Alliance manuscript #LSA-2023-02420-T

Lucy Dillon
School of Biological Sciences Queen's University Belfast

Dear Dr. Dillon,

Thank you for submitting your manuscript entitled "Accessory genes define species-specific pathways to antibiotic resistance." to Life Science Alliance. The manuscript was assessed by expert reviewers, whose comments are appended to this letter. We invite you to submit a revised manuscript addressing the Reviewer comments.

Thank you for this interesting contribution to Life Science Alliance. We are looking forward to receiving your revised manuscript.

Sincerely,

B. MANUSCRIPT ORGANIZATION AND FORMATTING:

Reviewer #1 (Comments to the Authors (Required)):

This paper addresses one of the issues that has plagued our understanding of AMR - the identification of the genes that are either directly involved in resistance or those genes that support the AMR phenotype.

This paper starts out with the observation that using a naive AMR phenotype detector - the kind of approach that is in common usage - results in a very poor prediction overall (~57%), with resistance to some antibiotics being very poorly predicted indeed.

They use multiple machine learning approaches to see if this would improve the identification of phenotypes. The answer is that they can produce huge improvements in prediction accuracy.

The example of COG0480 is particularly good - it makes it clear how AMR is achieved and that copy number sometimes matters in one direction and sometimes in the other.

I have read this paper several times and each time I found something new and interesting. It is a landmark paper in many ways and will be hugely influential in the ways in which we view antibiotic resistance and in how we tackle the problem.

I commend the authors for such a well-written, clear and important paper.

Reviewer #2 (Comments to the Authors (Required)):

The authors propose an interesting idea, which is being able to predict antibiotic resistance based on genomic data, and computational approaches, including machine learning. The data suggests that by using machine learning the accuracy of predicting antimicrobial resistance across 23 antibiotic models increases by more than 1.5.

The major comment I have regarding the manuscript is the writing. Sometimes, one needs to spend more than 5 minutes in a sentence to be able to fully understand what the authors try to say. So, for a revised version of the manuscript, it is really encouraged to improve on this.

Also, I could not find or see the supplementary material of the manuscript, is this available? or am I accidentally missing it?

Line 89: >50% of the models only predicted one phenotype, what does "one phenotype" refer to? and what would then multiple phenotypes correspond to in other models?

Line 113: same issue, what does this sentence mean: The AMR phenotypes were predicted accurately given that both phenotypes were distributed evenly in the data, what does both phenotypes refer to?

Line 117: same here, Severe imbalance in phenotypes in the training data

Lines 121-131: It is unclear to me how the authors reconcile the poor performance in Salmonella and Ciprofloxacin. The authors mention that in the model for Ampicillin the tree traversal showed that the majority of genomes were from Salmonella, so when excluded from the training, they would have been also excluded from the model, hence the testing is low. Would this mean that when you probe for a new organism absent in the training set the model will perform poorly? how does this affect how you think about the model? This brings me to the question of to what organisms do all the genomes belong to? How related are they (eg. Klebsiella, Salmonella, E. coli, are very closely related)? Also, how do you see strain variation within a species to influence your results?

The authors obtain two good prediction models, one based on AMR genes not thought to be involved in resistance to antibiotics, and the other based on eggNOG pathways. Both models perform very similarly. The authors point to the eggNOG model as being more informative. Could the authors elaborate on how they see which model could be more useful from a point of view of developing a drug towards a target? or to be applied in the clinics?

Queen's University Belfast
School of Biological Sciences
19 Chlorine Gardens
Belfast
BT9 5DL
Northern Ireland
01/12/2023

Dear Dr Eric Sawey,

Thank you for considering our manuscript for the Journal of Life Science Alliance. I would like to thank both reviewers for their feedback and comments. We have listed each comment in bold with our response.

Reviewer 1:

This paper addresses one of the issues that has plagued our understanding of AMR - the identification of the genes that are either directly involved in resistance or those genes that support the AMR phenotype.

This paper starts out with the observation that using a naïve AMR phenotype detector - the kind of approach that is in common usage - results in a very poor prediction overall (~57%), with resistance to some antibiotics being very poorly predicted indeed.

They use multiple machine learning approaches to see if this would improve the identification of phenotypes. The answer is that they can produce huge improvements in prediction accuracy.

The example of COG0480 is particularly good - it makes it clear how AMR is achieved and that copy number sometimes matters in one direction and sometimes in the other.

I have read this paper several times and each time I found something new and interesting. It is a landmark paper in many ways and will be hugely influential in the ways in which we view antibiotic resistance and in how we tackle the problem.

I commend the authors for such a well-written, clear and important paper.

We thank the reviewer 1 for their positive comments regarding our manuscript, we agree that COG0480 (tet(44)) provides a very good example of how machine learning approaches can identify key genes without any prior knowledge.

Reviewer 2: (Please use the new document to see the line numbers not the tracked changes document)

The authors propose an interesting idea, which is being able to predict antibiotic resistance based on genomic data, and computational approaches, including machine learning. The data suggests that by using machine learning the accuracy of predicting antimicrobial resistance across 23 antibiotic models increases by more than 1.5.

The major comment I have regarding the manuscript is the writing. Sometimes, one needs to spend more than 5 minutes in a sentence to be able to fully understand what the authors try to say. So, for a revised version of the manuscript, it is really encouraged to improve on this.

The major comment from reviewer 2 is the clarity of the sentences, we have addressed this issue by making the following changes:

- Had more consistency in the text i.e. we changed the wording from pathways to routes to avoid any confusion with biological pathways.

- Reduced and re-wrote the abstract to meet journal requirements
- Added clarity in the introduction (see lines 69-74, lines 91-150).
- Re-wrote the STRING protein-protein interaction section (“ML models identify putative additional antibiotic targets of AMR genes.”).
- Re-wrote the taxonomic analysis section (“Model accuracy is not reliant on specific taxonomy.”).
- Re-wrote the decision tree section (“Decision trees identify species-specific biological routes to resistance.”).
- Re-wrote the summary section.
- Overall we added a substantial amount of editing of the entire manuscript to increase clarity and readability (please see the tracked changes document for details).

Also, I could not find or see the supplementary material of the manuscript, is this available? or am I accidentally missing it?

The supplementary information is hosted on Open Science Framework at the following link:
<https://osf.io/CJ4BQ>.

This information was originally a view only link which was included in the manuscript in both the “Methods and Materials” and “Data Availability”, this has since been updated to the public link see lines 373 and 522. To view the supplementary content on OSF, find the section labelled “Files”, the files can be downloaded from the main page, or the user may view online by clicking the arrow in a box with a link to view all the supplementary information (in the file section).

Line 89: >50% of the models only predicted one phenotype, what does "one phenotype" refer to? and what would then multiple phenotypes correspond to in other models?

We have improved the clarity by adding “(either susceptible or resistance)” to the text. Multiple phenotypes just means that both phenotypes (susceptible and resistance) could both be predicted. This has been added to the text to improve clarity line 98.

Line 113: same issue, what does this sentence mean: The AMR phenotypes were predicted accurately given that both phenotypes were distributed evenly in the data, what does both phenotypes refer to?

We have added clarity to this, please see lines 127-131 to see the updated section: “This variation in performance across taxa may be due to an uneven distribution of susceptible vs resistant data in some of the training data, (for instance for testing Streptococcus AMR phenotype, there were 383 resistant genomes versus 1,807 susceptible genomes, compared to 2,364 resistant genomes versus 13 susceptible genomes for testing Salmonella AMR phenotype).”

Line 117: same here, Severe imbalance in phenotypes in the training data

We have improved the clarity by removing this and including it in the section stated above (see lines 127-131). Severe imbalance just refers to if there are more susceptible isolates than resistance isolates in the training data (or vice versa).

Lines 121-131: It is unclear to me how the authors reconcile the poor performance in Salmonella and Ciprofloxacin.

We have addressed this issue by re-writing the text please see lines 133 -141 for this new text "However, sometimes when the genus with the second-largest number of genomes for an antibiotic was excluded from the training data, the resulting models predicted their AMR phenotype poorly. For example, with Ampicillin when the second largest genus *Salmonella* (i.e. 1,048 *Salmonella* genomes vs 1,923 *Klebsiella* genomes) was excluded, the resulting model had an accuracy of only 34.3% when predicting the *Salmonella* AMR phenotypes. For Ciprofloxacin however, when *Salmonella* (which was the second largest genus in this model after *Klebsiella*) was excluded from the data, the resulting models had accuracy of 97.7% when predicting the *Salmonella* AMR phenotypes. This may be due to some genera being more closely related than others, such as *Klebsiella* and *Escherichia* compared to *Neisseria*".

The authors mention that in the model for Ampicillin the tree traversal showed that the majority of genomes were from Salmonella, so when excluded from the training, they would have been also excluded from the model, hence the testing is low. Would this mean that when you probe for a new organism absent in the training set the model will perform poorly? how does this affect how you think about the model?

We have addressed this by stating that these models are trained on mainly clinical data, therefore, would only recommend predicting the phenotype of other clinical isolates (see summary section line 359). However, we can see from our taxonomic analysis that we can predict species from distinct phyla groups from the training data well (i.e. *Streptococcus*, see line 125). Please also see Supplementary Table 3 in which this shows the results of the taxonomic analysis (remove one genus from training data and test the model with the excluded genus).

This brings me to the question of to what organisms do all the genomes belong to?

There are 19 unique genera used in this study, can be seen in the introduction (line 69). The specific numbers can be found in the supplementary table 13 or the supplementary table 3 which has the number of genomes from each genus for each antibiotic model.

How related are they (eg. Klebsiella, Salmonella, E. coli, are very closely related)?

The majority of the genomes are from the Pseudomonadota phyla, specifically from clinical isolates. Nevertheless, there are strains from other phyla such as Bacillota and Actinomycetota.

The list of genomes used in this study is available on the GitHub:

https://github.com/LucyDillon/AMR_ML_paper/tree/main if the reader wants to investigate the distribution of the genomes. The numbers of each genus are reported in Supplementary Table 3 and Supplementary Table 13.

Also, how do you see strain variation within a species to influence your results?

We did not look into individual strain variation, yet the tree traversal results do show that closely related taxonomy do follow many of the same pathways to resistance. We also found that same species can take alternative routes to resistance which could be an indicator of variation within the strains. Supplementary Figure 6. For example, in Supplementary Figure 6, part U (Tetracycline tree traversals), this shows that there are 7 different routes to AMR phenotype for *Neisseria gonorrhoeae*, which could indicate variation within the species.

The authors obtain two good prediction models, one based on AMR genes not thought to be involved in resistance to antibiotics, and the other based on eggNOG pathways. Both models

Queen's University Belfast
School of Biological Sciences
19 Chlorine Gardens
Belfast
BT9 5DL
Northern Ireland
01/12/2023

perform very similarly. The authors point to the eggNOG model as being more informative. Could the authors elaborate on how they see which model could be more useful from a point of view of developing a drug towards a target? or to be applied in the clinics?

To clarify for the reviewers understanding, within this study we made decision tree models using defined AMR genes (as stated in the CARD database) and additional models which use eggNOG gene families. The final comment refers to elaborating on which model we as having the most potential from a clinical and drug development viewpoint. We have tried to address this by adding the following line in our Summary section line 351 "Additionally, we demonstrate that incorporation all genes from a genome (and not just the known AMR genes) allows the identification of species-specific routes to antibiotic resistance, involving genes not previously identified as involved in this process." and line 362 "Lastly, the accessory genes identified as involved in AMR phenotype could also provide novel drug targets to combat AMR."

We are grateful for the comments provided by reviewer 2, we hope these changes will be satisfactory.

Yours Sincerely,
Lucy Dillon, on behalf of all authors.

December 28, 2023

RE: Life Science Alliance Manuscript #LSA-2023-02420-TR

Ms. Lucy Dillon
Queen's University Belfast
School of Biological Sciences
19 Chlorine Gardens
Belfast BT9 5DL
United Kingdom

Dear Dr. Dillon,

Thank you for submitting your revised manuscript entitled "Accessory genes define species-specific routes to antibiotic resistance.". We would be happy to publish your paper in Life Science Alliance pending final revisions necessary to meet our formatting guidelines.

- please remove the figures from the manuscript file and insert their legends after the references section
- please use the [10 author names et al.] format in your references (i.e., limit the author names to the first 10)
- please add a callout for Figures 1A-C and tables S2, S3, S7-S10, S12-S14 to your main manuscript text

A. FINAL FILES:

B. MANUSCRIPT ORGANIZATION AND FORMATTING:

Sincerely,

January 3, 2024

RE: Life Science Alliance Manuscript #LSA-2023-02420-TRR

Ms. Lucy Dillon
Queen's University Belfast
School of Biological Sciences
19 Chlorine Gardens
Belfast BT9 5DL
United Kingdom

Dear Dr. Dillon,

Thank you for submitting your Research Article entitled "Accessory genes define species-specific routes to antibiotic resistance.". It is a pleasure to let you know that your manuscript is now accepted for publication in Life Science Alliance. Congratulations on this interesting work.

DISTRIBUTION OF MATERIALS:

Again, congratulations on a very nice paper. I hope you found the review process to be constructive and are pleased with how the manuscript was handled editorially. We look forward to future exciting submissions from your lab.

Sincerely,
